# Sn-Substituted Argyrodite Li_6_PS_5_Cl Solid Electrolyte for Improving Interfacial and Atmospheric Stability

**DOI:** 10.3390/ma16072751

**Published:** 2023-03-29

**Authors:** Seul-Gi Kang, Dae-Hyun Kim, Bo-Joong Kim, Chang-Bun Yoon

**Affiliations:** Department of Advanced Materials Engineering, Tech University of Korea, Siheung-si 15073, Republic of Korea

**Keywords:** solid electrolyte, argyrodite, liquid-phase synthesis, air stability, Sn (IV) substitution

## Abstract

Sulfide-based solid electrolytes exhibit good formability and superior ionic conductivity. However, these electrolytes can react with atmospheric moisture to generate H_2_S gas, resulting in performance degradation. In this study, we attempted to improve the stability of the interface between Li metal and an argyrodite Li_6_Ps_5_Cl solid electrolyte by partially substituting P with Sn to form an Sn–S bond. The solid electrolyte was synthesized via liquid synthesis instead of the conventional mechanical milling method. X-ray diffraction analyses confirmed that solid electrolytes have an argyrodite structure and peak shift occurs as substitution increases. Scanning electron microscopy and energy-dispersive X-ray spectroscopy analyses confirmed that the particle size gradually increased, and the components were evenly distributed. Moreover, electrochemical impedance spectroscopy and DC cycling confirmed that the ionic conductivity decreased slightly but that the cycling behavior was stable for about 500 h at X = 0.05. The amount of H_2_S gas generated when the solid electrolyte is exposed to moisture was measured using a gas sensor. Stability against atmospheric moisture was improved. In conclusion, liquid-phase synthesis could be applied for the large-scale production of argyrodite-based ${\mathrm{Li}}_{6}{\mathrm{PS}}_{5}\mathrm{Cl}$ solid electrolytes. Moreover, Sn substitution improved the electrochemical stability of the solid electrolyte.

## 1. Introduction

With the increasing interest in environmental issues such as climate change and environmental pollution, the applications of Li-ion batteries are expanding from the small battery market for electronic devices to the medium and large battery markets for electric vehicles and energy storage systems. Moreover, the power and energy density requirements for future batteries are increasing with the continuous growth in energy storage demand [1,2]. In conventional Li-ion batteries, the dense packing and narrow spacing of electrodes increase the energy density of the battery; however, these batteries also exhibit a higher risk of explosion. When a liquid electrolyte, which is a flammable material, is used, the risk of ignition and explosion due to thermal runaway increases with the increase in internal battery temperature that is caused by continuous heat generation [3,4]. In fact, explosions and fire accidents have been consistently reported in recent years. With the expanding Li-ion battery market and growing interest in safety issues, researchers are conducting various studies to improve safety [5].

An all-solid-state battery has all solid components, which is achieved by replacing the flammable liquid electrolyte that is typically used in traditional batteries with a nonflammable inorganic solid electrolyte. Solid electrolytes are more resistant to shock and vibration and can operate over a wider range of temperatures [6]. Several important requirements must be satisfied before applying solid electrolytes to all-solid-state batteries, including high ionic conductivity, high chemical compatibility with electrodes, wide electrochemical window, low electronic conductivity, high energy density and power, and sufficient mechanical strength [7,8]. Although solid electrolytes are expected to improve the safety and energy density of batteries when compared to liquid electrolytes, they still exhibit issues such as low ionic conductivity, low solid–solid contact with electrode materials (interface resistance), and poor mobility owing to their fragility. Large polarization is induced by slow Li-ion diffusion through the interface between the bulk electrolyte and the electrodes. As poor contact between the solid electrolytes and Li metal can cause high interface resistance and short battery life, various studies to improve ionic conductivity and solid–solid contact are underway [9].

Some sulfide-based solid electrolytes have been shown to exhibit lower activation energies and higher ionic conductivities than oxide- and polymer-based electrolytes due to the body-centered cubic anion sublattice favoring the movement of lithium ions [10,11,12]. Sulfide-based electrolytes have considerably high ionic conductivity ($10^{-2}$ to $10^{-4}$ S/cm) at room temperature. When compared to oxide-based solid electrolytes, they have a lower modulus of elasticity; therefore, contact with the active material can be improved through mechanical compression, and the density can be increased because of their superior formability [13,14]. However, owing to their low electrochemical stability, sulfide-based electrolytes can react with moisture in the air generating H_2_S gas and leading to performance degradation [15,16,17].

In this study, in a variation from the solid-phase synthesis method, which has a long synthesis time and exhibits limitations in mass production, solid electrolytes were synthesized using a liquid-phase synthesis method, thereby shortening the process time and procedure. The liquid-phase process has diverse applications, such as the ability to maximize the contact surface between the electrolyte and electrodes by producing sheet-type electrodes [18,19,20]. 

Researchers have performed various studies in recent years to improve ionic conductivity and chemical stability, categorized into pseudobinary (i.e., ${\mathrm{Li}}_{2}S–P_{2}S_{5}$ or ${\mathrm{Li}}_{2}S–{\mathrm{MS}}_{2}$, M = Ge, Si, Sn), pseudoternary (i.e., ${\mathrm{Li}}_{2}S–P_{2}S_{5}–{\mathrm{MS}}_{2}$ or, ${\mathrm{Li}}_{2}S$–$P_{2}S_{5}$–LiX, M = Ge, Si, Sn, etc.; X = F, Cl, Br, and I), and pseudo-quaternary systems (${\mathrm{Li}}_{2}S$–$P_{2}S_{5}$–${\mathrm{MS}}_{2}$–LiX, M = Ge, Si, Sn, etc.; X = F, Cl, Br, and I). Among them, in particular, the ${\mathrm{Li}}_{2}S$–$P_{2}S_{5}$–${\mathrm{MS}}_{2}$–LiX (M = Ge, Si, Sn, etc.; X = F, Cl, Br, I) system exhibited high ionic conductivity [21,22,23]. There are several methods to improve stability; for example, the in situ ALD passivation of solid-state Li electrolyte and the substitution of elements based on the hard–soft acid–base theory [24,25]. In this study, elemental substitution was used. Based on this composition, we investigated how doping with Sn material influences the properties of a solid electrolyte by partially substituting P_2_S_5_ in the argyrodite ${\mathrm{Li}}_{6}{\mathrm{PS}}_{5}\mathrm{Cl}$ (LPSCl) solid electrolyte with SnS_2_. There are studies on substituting Sn for P in LPSI or elements such as O in LPSCl, but there are no reports on substituting Sn, which has a large atomic radius, in LPSCl.

## 2. Materials and Methods

### 2.1. Liquid-Phase Synthesis of the $Li_{6+x}P_{1-x}Sn_{x}S_{5}Cl\;$ Solid Electrolyte

In this study, sulfide-based solid electrolytes were synthesized using the liquid-phase method. ${\mathrm{Li}}_{2}S$ (99.9%, Alfa Aesar, Seoul, Korea), $P_{2}S_{5}$ (99%, Sigma-Aldrich, Seoul, Korea), LiCl (99.9%, Kojundo Korea, Uiwang, Korea), and ${\mathrm{SnS}}_{2}$ (99.9%, Kojundo Korea, Uiwang, Korea) were used as raw materials. Tetrahydrofuran (THF; 99.8%, anhydrous, Alfa Aesar, Seoul, Korea) and ethyl alcohol (EtOH; 99.9%, anhydrous, Daejung Chemicals & Metals Co., Ltd., Siheung, Korea) were used for the solutions. Argyrodite LPSCl was used as the basic composition of the solid electrolyte, and the amount of ${\mathrm{SnS}}_{2}$ added was adjusted. The corresponding chemical reaction equation can be written as: (5 + x) ${\mathrm{Li}}_{2}S$ + (1 − x) $P_{2}S_{5}$ + (2x) ${\mathrm{SnS}}_{2}$ + 2LiCl = $2{\;\mathrm{Li}}_{6+x}P_{1-x}{\mathrm{Sn}}_{x}S_{5}\mathrm{Cl}$, where x is the substitution content of Sn (0 ≤ x ≤ 0.1).$\;{\mathrm{Li}}_{6+x}P_{1-x}{\mathrm{Sn}}_{x}S_{5}\mathrm{Cl}\;$(x = 0, 0.025, 0.05, 0.075, 0.1) solid electrolytes were synthesized via a step-by-step process as follows. In a glass vial, the raw materials ${\mathrm{Li}}_{2}S$, $P_{2}S_{5}$, and ${\mathrm{SnS}}_{2}$ were mixed in the THF solvent in the ratio of 3 + x:1 − x:2x using a magnetic stirrer bar for 12 h to obtain a THF suspension. Simultaneously, in another glass vial, ${\mathrm{Li}}_{2}S$ and LiCl were dissolved in EtOH in 1:1 ratio and stirred for 12 h at 25 °C. After stirring, the THF suspension and EtOH solution were mixed to obtain a THF-EtOH precursor solution of LPSCl-xSn at 25 °C. To evaporate the solvent used in synthesis, we dried the mixed solution in a vacuum chamber for 4 h at 120 °C to obtain a solid powder. To improve the crystallinity of the solid powder, we heat-treated it in a quartz crucible for 6 h at 550 °C. All synthesis processes were conducted in a glove box under an inert Ar atmosphere to minimize the contact between the sulfide-based solid electrolyte and air. Figure 1 is a schematic diagram of the liquid-phase synthesis process used for synthesizing the solid electrolyte.

### 2.2. Characteristic Analysis of $Li_{6+x}P_{1-x}Sn_{x}S_{5}Cl$ Solid-Electrolyte Materials

An X-ray diffractometer (D2 PHASER, Bruker, Billerica, MA, USA) was used to analyze the crystal structure of the synthesized solid electrolyte. To prevent the sulfide-based solid electrolyte from reacting with atmospheric moisture and decomposing during measurement, we pre-sealed the specimen using a polyimide film with high transmittance in a glove box, thus preventing exposure to air during the analysis process. A Cu tube (λ = 1.54184) was used to perform measurements at a rate of 1° per min for 2θ ranging from 10° to 80°.

The surface morphology of the solid-electrolyte powder was observed in vacuum using a field-emission scanning electron microscope (Nova NanoSEM 450, FEI, Hillsboro, OR, USA). Furthermore, to identify the elemental distribution of the solid electrolyte, elemental analysis was conducted via energy-dispersive X-ray spectroscopy (EDS, FEI, Hillsboro, OR, USA). A mapping analysis was conducted for the constituent elements of the sulfide-based solid electrolyte, i.e., S, P, Cl, and Sn.

To confirm the degree of reaction of the synthesized solid electrolyte to atmospheric moisture, a gas meter (GasTiger 2000, Wandi, Gunpo, Korea) was used to measure the amount of H_2_S gas produced by the solid electrolyte when exposed to air. In the experiment, 0.1 g of the solid-electrolyte powder and the gas sensor were placed in a sealed case maintained at a relative humidity of 25% and indoor temperature of 25 °C; then, the amount of H_2_S gas generated for 15 min was measured.

### 2.3. Evaluation of the Electrochemical Properties of Li_6 + x_P_1 − x_Sn_x_S_5_Cl Solid Electrolyte 

To measure the Li ionic conductivity of the solid electrolytes, we prepared circular pellets by uniaxially pressing the solid-electrolyte powder at 25 MPa using a 10 mm diameter mold. The thickness of the prepared pellets was first measured; subsequently, In foils with a diameter of 8 mm were punched and attached to both sides of the pellets and then vacuum-packed for the cold isostatic pressing (CIP) process. To improve the electrolyte–electrode contact and density of the solid-electrolyte pellets, we pressure-molded these pellets at 250 MPa for 5 min using a CIP device (SCIP50150-3 kB, Samyang Ceratech, Incheon, Korea). A half-cell was then assembled using a 2032 coin-type cell. Electrochemical impedance spectroscopy (EIS; VersaSTAT 3, Princeton Applied Research, Tennessee, USA) was used to measure the impedance of the assembled cell at 25 °C. The EIS measurements were conducted at an amplitude of 10 mV and frequency range of 0.1 to 1 MHz.

The ionic conductivity was measured in the temperature range of 25 to 75 °C to calculate the activation energy. The equation to calculate the activation energy (Ea) is as follows. Here, kB is the Boltzmann constant and T is the absolute temperature. After measuring the ionic conductivity at various temperatures, Ea was calculated from the slope of the Arrhenius plot.
(1)$$\sigma =\sigma_{0}exp\;\left[\frac{-\Delta E}{k_{B}T}\right]$$

To perform direct current (DC) cycling measurements on the solid electrolytes, we prepared circular pellets by uniaxially pressing the solid-electrolyte powder at 25 MPa using a 10 mm diameter mold. Li metal discs with a diameter of 9 mm were punched and attached to both sides of the prepared pellets; subsequently, the specimen was vacuum-packed and then pressure-molded at 250 Mpa for 5 min using the CIP device. A half-cell was then assembled using a 2032 coin-type cell. A battery charge/discharge tester (WBCS 3000 Cycler, WonATech, Seoul, Korea) was used to perform DC cycling measurements on the assembled coin cell. We performed 215 cycles (approximately 500 h) at a current density of 0.1 mA/cm^2^ and cut-off charge and discharge cycles at 1-h intervals to evaluate the long-term stability between the Li metal and the solid electrolyte. Excluding the CIP process, the coin cell preparation process for ionic conductivity and DC cycling measurements was performed in a glove box with an Ar atmosphere to minimize the reactions of the sulfide-based solid electrolyte with atmospheric moisture.

## 3. Results and Discussion

### 3.1. Synthesis of Solid Electrolyte and Structural Characterization

Figure 2a shows the X-ray diffraction (XRD) analysis results for each component of the synthesized argyrodite L_6 + x_P_1 − x_Sn_x_S_5_Cl solid electrolyte. The results show that the argyrodite structure was maintained in all compositions of the synthesized solid electrolytes (x = 0 to 0.1). However, as the substitution ratio of SnS_2_ increased, peaks of unreacted Li_2_S, LiCl, and Li_3_PO_4_ were detected, and their intensities also increased. This signifies that the excessive substitution of SnS_2_ increases the generation of impurities. These impurities impact the grain-boundary resistance of the solid-electrolyte pellet and can potentially degrade its performance [19]. Figure 2b shows the magnified XRD patterns in the range of 23° to 34°, indicating a right shift of the XRD peaks as the substitution ratio of SnS_2_ increases. As the P atoms (r = 1.1 Å) are substituted by the Sn atoms (r = 1.4 Å), which have a large ionic radius, the substitution ratio of SnS_2_ increases, and the crystal structure is deformed, causing the XRD peaks to shift.

### 3.2. Solid-Electrolyte Shape Observation and Elemental Analysis

Figure 3 shows the scanning electron microscopy observations of the solid electrolytes synthesized via the liquid-phase synthesis method. Figure 3a–e show the powder surface observations when the SnS_2_ substitution ratio is x = 0, 0.025, 0.05, 0.075, and 0.1, respectively. The results indicate that as the SnS_2_ substitution ratio increases, the particles gradually increase in size due to the low melting point of SnS_2_.

Figure 4 illustrates the EDS mapping analysis results of the solid electrolytes synthesized using the liquid-phase synthesis method. Figure 4a shows the analysis results of elemental carbon, which was analyzed to check for residual THF and EtOH. Some amount of carbon was detected because of the carbon tape used to fix the sample; however, no carbon was detected in the solid-electrolyte particles. This indicates that the solvents were completely removed. Figure 4b shows the elemental analysis results of S, P, and Sn, the constituent elements of the solid electrolyte, with an Sn substitution ratio of x = 0.025. The results show that S, P, and Sn are evenly distributed throughout the LPSCl-Sn solid electrolyte. The solid electrolyte was successfully synthesized without a concentration gradient of constituent elements through the liquid phase synthesis method.

### 3.3. Evaluation of the Electrochemical Properties of the Solid Electrolyte

Figure 5 illustrates the measurements of Li ionic conductivity at 25 °C through an EIS analysis of the LPSCl-xSn solid electrolytes according to the substitution ratio. The maximum ionic conductivity of 9.27 × 10^−4^ S/cm s was observed at x = 0, and as the SnS_2_ substitution ratio increases, the Li ionic conductivity marginally decreases to a minimum of 5.36 × 10^−4^ S/cm at x = 0.1. We speculated that, as the substitution ratio of SnS_2_ increased, the unreacted leached impurities in the solid electrolytes resulted in a decrease in ionic conductivity.

Figure 6 shows the changes in activation energy calculated through the Arrhenius plot and the slope calculated based on the measured ionic conductivity corresponding to the change in temperature from 25 to 75 °C. As the SnS_2_ substitution ratio increases, the activation energy of the solid electrolytes gradually decreases from 0.285 eV at x = 0 to a minimum of 0.237 eV at x = 0.075. However, for x = 0.1, the activation energy increases to 0.252 eV. We judged that Sn, which has a large atomic radius, expands the Li-ion movement channels and reduces the activation energy to a certain degree; however, when excessive amounts of Sn are added, the lattice structure is severely deformed and the activation energy increases. The activation energy and ionic conductivity show different trends, indicating the influence of a complex mechanism.

### 3.4. Evaluation of Electrochemical Stability of Solid Electrolytes

Figure 7 shows the DC cycling measurements of the Li–Li symmetric cell to evaluate the long-term interfacial stability of the Li metal and solid electrolyte. In the LPSCl (x = 0) solid electrolyte, the driving voltage gradually increased in later cycles when compared to the early cycles, and the voltage curve became unstable after approximately 250 h. In contrast, in the solid electrolyte with x = 0.05, a low voltage was maintained over a long period, and the voltage did not increase from the initial cycles to the later cycles but remained stable. The most stable behavior was shown at x = 0.05. Since the intense fluctuations and gradually decreasing overvoltage observed for x = 0 indicate poor Li plating and stripping at the Li metal and LPSCl interface [23], these results confirm that an appropriate substitution ratio of SnS_2_ improves the interfacial stability between the Li metal and the electrolyte.

Figure 8 shows the gas sensor measurements of the amount of H_2_S gas that was generated when the Li_6 + x_P_1 − x_Sn_x_S_5_Cl solid electrolyte reacted with atmospheric moisture. The experiment was performed to determine the atmospheric stability of the SnS_2_-substituted solid electrolyte under conditions of 25% relative humidity at 25 °C. The results indicate that, as the SnS_2_ substitution ratio increased, the amount of H_2_S gas generated gradually decreased and then increased at x = 0.1. This is because when SnS_2_ is excessively added, a large number of unsynthesized Li_2_S, LiCl, etc., are present and become rather unstable, leading to deterioration in properties. The LPSCl solid electrolyte with the conventional composition generated 127 ppm of H_2_S gas after 15 min of exposure to air. The solid electrolyte with x = 0.075 generated 92 ppm of H_2_S gas after 15 min of exposure to air, which was the smallest amount observed in the experiment. These results demonstrate that SnS_2_ substitution is effective to a certain degree in suppressing the reactivity of the solid electrolytes to atmospheric moisture.

## 4. Conclusions

In this study, sulfide-based LPSCl solid electrolytes were synthesized step-by-step using a liquid-phase synthesis method based on THF and EtOH solvents, thereby reducing the synthesis time and simplifying the process.

As the SnS_2_ substitution ratio increased, the Li ionic conductivity of the L_6 + x_P_1 − x_Sn_x_S_5_Cl solid electrolyte decreased; however, this also caused the activation energy calculated through the Arrhenius plot to decrease. This phenomenon is attributed to the influence of a complex mechanism.

The solid electrolyte with an SnS_2_ substitution ratio of x = 0.05 exhibited stable Li metal plating and stripping behavior for approximately 500 h, as well as improved electrochemical stability.

Furthermore, to evaluate the reactivity of the solid electrolyte to atmospheric moisture, the amount of H_2_S gas that was generated when the electrolyte was exposed to air was measured using a gas sensor. The measurements indicated that the amount of H_2_S gas generated decreased as the SnS_2_ substitution ratio increased.

The findings of the study confirmed that SnS_2_ substitution forms strong bonds between Sn and S, preventing the solid electrolyte from chemically decomposing and improving its chemical stability. Liquid-phase synthesis of solid electrolytes can be applied to produce composite electrodes using a homogeneous solid electrolyte solution, and the overall battery performance can be improved by applying composite electrodes to all-solid-state batteries. Through further research, it will be possible to contribute to the mass production of sulfide-based solid electrolytes by improving the stability of sulfide systems that are vulnerable to moisture.

## Figures and Tables

**Figure 1 materials-16-02751-f001:**
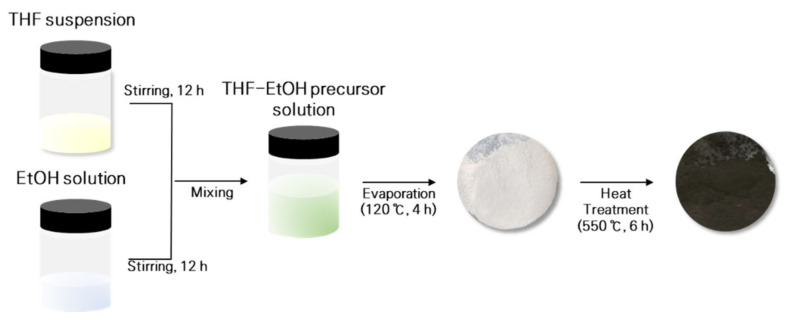
Schematic diagram of the liquid-phase synthesis process of solid electrolyte.

**Figure 2 materials-16-02751-f002:**
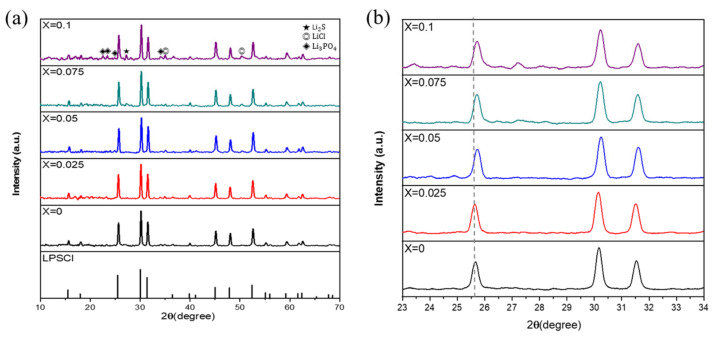
(**a**) X-ray diffraction (XRD) patterns of LPSCl-xSn with different amounts of Sn substitution (x = 0, 0.025, 0.05, 0.075, 0.1) (**b**) Shift in the XRD patterns (23° < 2θ < 34°).

**Figure 3 materials-16-02751-f003:**
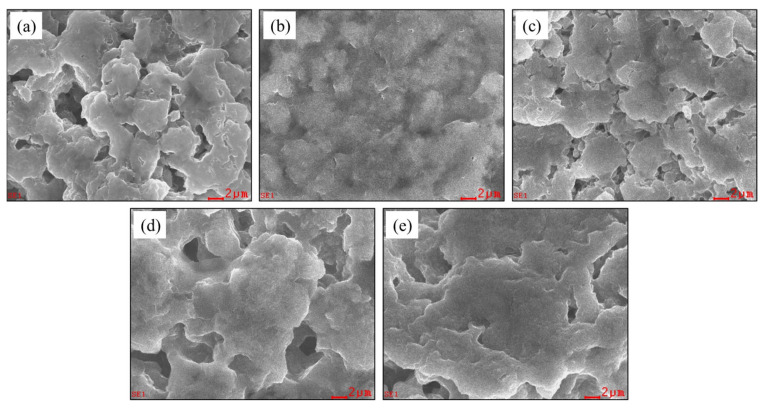
Scanning electron microscopy images of LPSCl-xSn sulfide-based electrolytes with different amounts of Sn substitution (**a**) x = 0, (**b**) x = 0.025, (**c**) x = 0.05, (**d**) x = 0.075, (**e**) x = 0.1.

**Figure 4 materials-16-02751-f004:**
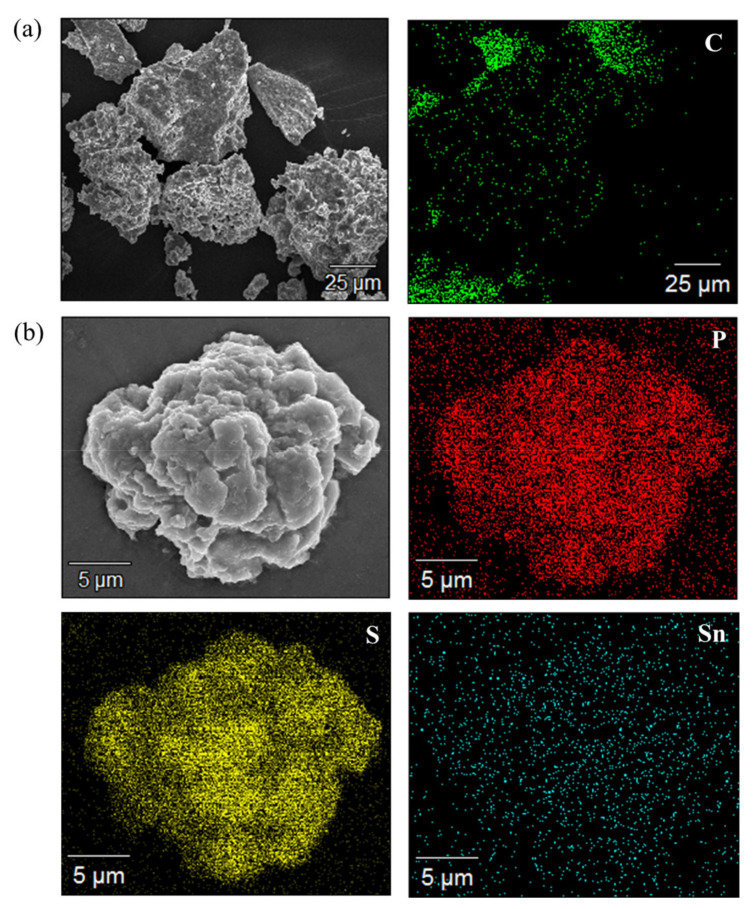
Energy-dispersive X-ray spectroscopy images of LPSCl-xSn sulfide-based electrolytes (**a**) x = 0 (**b**) x = 0.025.

**Figure 5 materials-16-02751-f005:**
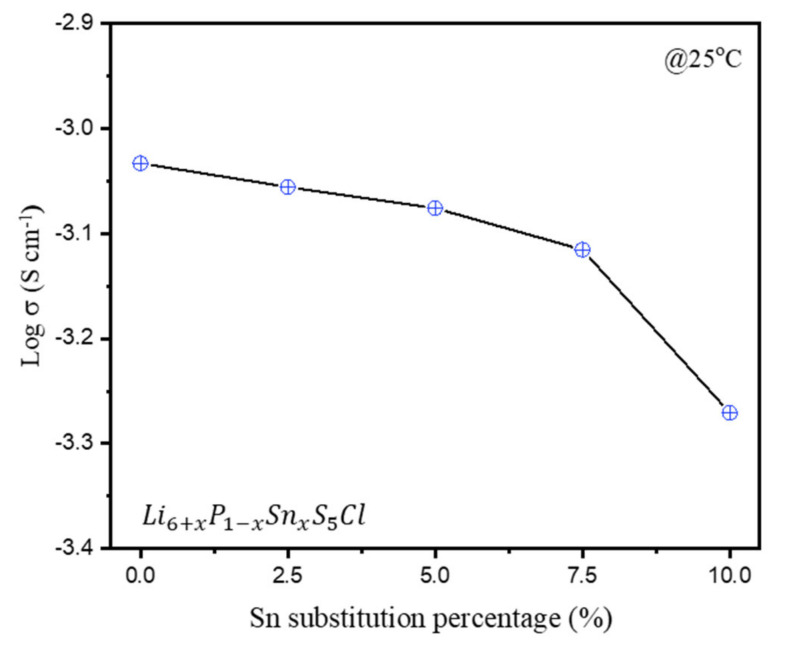
Ionic conductivity of LPSCl-xSn sulfide-based electrolytes with different amounts of Sn substitution (x = 0, 0.025, 0.05, 0.075, 0.1).

**Figure 6 materials-16-02751-f006:**
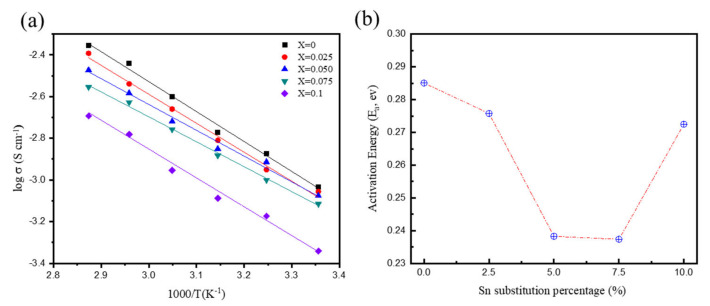
(**a**) Arrhenius plots of the LPSCl-xSn sulfide-based electrolytes with different amounts of Sn substitution; (**b**) Change trend of the activation energy against different substitution percentages.

**Figure 7 materials-16-02751-f007:**
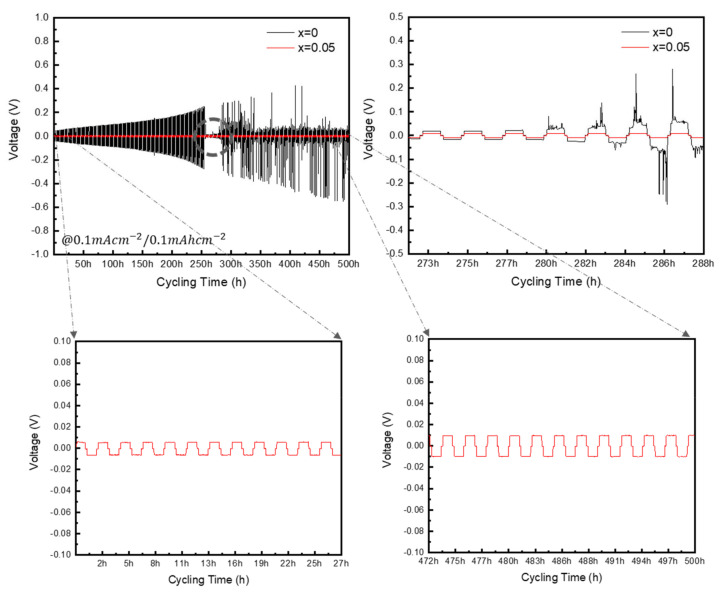
Li–Li symmetric cells performance of LPSCl-xSn sulfide-based electrolytes with different amounts of Sn substitution (x = 0, x = 0.05).

**Figure 8 materials-16-02751-f008:**
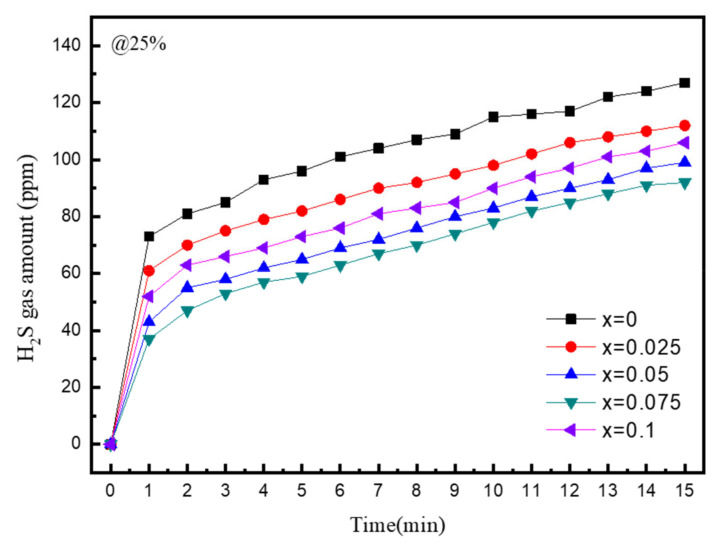
$H_{2}S$ gas emission amounts when LPSCl-xSn sulfide electrolytes with different Sn substitution amounts (x = 0, 0.025, x = 0.05, 0.075, 0.1) are exposed to moisture in the atmosphere.

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
