# Peer review of "Sn-Substituted Argyrodite Li6PS5Cl Solid Electrolyte for Improving Interfacial and Atmospheric Stability"

_materials, 2023, doi:10.3390/ma16072751_

Round 1
Reviewer 1 Report
Comments on the paper «Sn-substituted Argyrodite ???????? solid electrolyte for improving interfacial and atmospheric stability» by Seulgi Kang, Daehyun Kim, Bojoong Kim, and Chang-Bun Yoon submitted to Materials.
Lithium-ion batteries play an important role in the life quality of modern society as the dominant technology for use in portable electronic devices such as mobile phones, tablets and laptops. Sulfide-based electrolytes have considerably high ionic conductivity at room temperature. However, owing to their low electrochemical stability, sulfide-based electrolytes can react with moisture in the air generating H2S gas and leading to performance degradation. The presented work investigated the effect of Sn entering (as SnS2) the sulfide-based LPSCl solid electrolytes on the Li ionic conductivity and the amount of H2S gas was generated. Solid electrolytes were synthesized using a liquid-phase synthesis method, thereby shortening the process time and procedure. The liquid-phase process has diverse applications, such as the ability to maximize the contact surface between the electrolyte and electrodes by producing sheet type electrodes.
The topic interesting and original.
Analysis methods are as follows: a) an X-ray diffractometer (D2 PHASER, Bruker, MA, US) was used to analyze the crystal structure of the synthesized solid electrolyte; b) the surface morphology of the solid-electrolyte powder was observed in vacuum using a field-emission scanning electron microscope (Nova NanoSEM 450, FEI, US); c) to confirm the degree of reaction of the synthesized solid electrolyte to atmospheric moisture, a gas meter (GasTiger 2000, Wandi, Korea) was used to measure the amount of H2S gas produced by the solid electrolyte when exposed to air.
I do not hesitate to state that the paper is based on a detailed various methods study. The results are important for the mass production of sulfide-based solid electrolytes.
The work is quite interesting. I have no objection on the data interpretation. Further are specific comments:
1) in the article the authors indicate about partially substituting P with Sn to form an Sn–S bond. However, the substitution reaction is not given, nor is there a reference to the formation of such structures.
2) “The results indicate that, as the SnS2 substitution ratio increased, the amount of H2S gas
generated gradually decreased and then increased at x=0.1.” The text does not explain very well what is the reason for this increase in H2S gas at x = 0.1. And this is an important question that needs to be understood when using SnS2 in an electrolyte.
Some typos in the text were changed in the file ‘materials-2285116-peer-review-v1_1.pdf’
In conclusion, the paper is suitable for publication in Materials after minor-to-moderate revision.
Author Response
Dear Editor:
I am pleased to re-submit our revised version of “Sn-substituted Argyrodite Li6PS5Cl solid electrolyte for improving interfacial and atmospheric stability” for publication. I appreciated the thorough review and constructive criticisms of the reviewers. I have addressed each of their concerns below and have rewritten sections of the paper to provide clarity. I hope the revision has improved the paper to a level of the reviewers’ satisfaction.
Reviewers' comments:
I do not hesitate to state that the paper is based on a detailed various methods study. The results are important for the mass production of sulfide-based solid electrolytes.
The work is quite interesting. I have no objection on the data interpretation. Further are specific comments:
- in the article the authors indicate about partially substituting P with Sn to form an Sn–S bond. However, the substitution reaction is not given, nor is there a reference to the formation of such structures.
- Relevant reaction equations have been inserted into the manuscript.
- (5+x) + (1−x) + (2x) + =
- The revised part of the manuscript is marked in blue. (2 of 11 page, 2. Materials and Methods)
2) “The results indicate that, as the SnS2 substitution ratio increased, the amount of H2S gas
generated gradually decreased and then increased at x=0.1.” The text does not explain very well what is the reason for this increase in H2S gas at x = 0.1. And this is an important question that needs to be understood when using SnS2 in an electrolyte.
- This is because when SnS2 is excessively added, a large number of unsynthesized Li2S, LiCl etc. are present and become rather unstable, leading to deterioration in properties.
- The revised part of the manuscript is marked in blue. (8 of 11 page)
Some typos in the text were changed in the file ‘materials-2285116-peer-review-v1_1.pdf’
In conclusion, the paper is suitable for publication in Materials after minor-to-moderate revision.
Reviewer 2 Report
This manuscript studied the Li ionic conductivity variation of the L6+xP1-xSnxS5Cl solid electrolyte with the dosage of SnS2 substitution precursor. The solid electrolyte with a SnS2 substitution ratio of x=0.05 demonstrated consistent Li metal plating and stripping behavior over 500 h, as well as increased electrochemical stability. However, to improve the current manuscript, I believe the following comments should be addressed:
Comments 1): At the end of the introduction, the ??2?–?2?5–??2–??? is introduced abruptly. Why is this system widely studied? Why is effort spent to improve this material system? Please elaborate.
Comment 2): References are missing for the introduction. For example, lines 7-11 in the first paragraph, etc. Any similar issues should be avoided.
Comments 3): In Figure 3, “almost no voids are observed at a substitution ratio of x=0.025, and the size of the voids gradually increases at higher substitution ratios.” Why 0.025 doping can heal the voids? Please elaborate.
Comments 4): The authors have studied the elemental distribution of the LPSCl-xSn. Does the elemental percentage match the ideal composition?
Comments 5): It would benefit the public if the authors could compare the stabilization methods. For example, the in-situ ALD passivation of solid-state Li electrolyte. See Wang et al., Surfaces and Interfaces 33 (2022) 102280 (https://doi.org/10.1016/j.surfin.2022.102280)
Author Response
Dear Editor:
I am pleased to re-submit our revised version of “Sn-substituted Argyrodite Li6PS5Cl solid electrolyte for improving interfacial and atmospheric stability” for publication. I appreciated the thorough review and constructive criticisms of the reviewers. I have addressed each of their concerns below and have rewritten sections of the paper to provide clarity. I hope the revision has improved the paper to a level of the reviewers’ satisfaction.
Reviewer #2:
This manuscript studied the Li ionic conductivity variation of the L6+xP1-xSnxS5Cl solid electrolyte with the dosage of SnS2 substitution precursor. The solid electrolyte with a SnS2 substitution ratio of x=0.05 demonstrated consistent Li metal plating and stripping behavior over 500 h, as well as increased electrochemical stability. However, to improve the current manuscript, I believe the following comments should be addressed:
Comments 1): At the end of the introduction, the ??2?–?2?5–??2–??? is introduced abruptly. Why is this system widely studied? Why is effort spent to improve this material system? Please elaborate.
- According to one study, pseudo-quaternary has generally higher ionic conductivity and improved stability compared to other systems. (reference number 21)
- Pseudobinary, pseudoternary and pseudo-quaternary systems
- An additional explanation was added to the manuscript. (2 of 11 page)
Comment 2): References are missing for the introduction. For example, lines 7-11 in the first paragraph, etc. Any similar issues should be avoided.
- References have been modified in the manuscript.
Comments 3): In Figure 3, “almost no voids are observed at a substitution ratio of x=0.025, and the size of the voids gradually increases at higher substitution ratios.” Why 0.025 doping can heal the voids? Please elaborate.
- I deleted the related content because there seems to be room for misunderstanding.
Comments 4): The authors have studied the elemental distribution of the LPSCl-xSn. Does the elemental percentage match the ideal composition?
- The EDS component analysis was to check whether the solvent was well dried in the solid electrolyte synthesized through the liquid phase synthesis method and whether there was no aggregation phenomenon.
- I added it because it seemed that the explanation was insufficient in the manuscript. (6 of 11 page)
Comments 5): It would benefit the public if the authors could compare the stabilization methods. For example, the in-situ ALD passivation of solid-state Li electrolyte. See Wang et al., Surfaces and Interfaces 33 (2022) 102280 (https://doi.org/10.1016/j.surfin.2022.102280)
- In the manuscript, ALD passivation was introduced as one of the stabilization methods. (2 of 11)
.